# Cardiac Metabolism in Sepsis

**DOI:** 10.3390/metabo11120846

**Published:** 2021-12-06

**Authors:** Satoshi Kawaguchi, Motoi Okada

**Affiliations:** 1Department of Anatomy, Cell Biology and Physiology, Indiana University School of Medicine, Bloomington, IN 46202, USA; s-kawa@asahikawa-med.ac.jp; 2Department of Emergency Medicine, Asahikawa Medical University, Asahikawa 078-8510, Japan

**Keywords:** sepsis, SICM, β-adrenergic receptor, metabolic switch, ATP

## Abstract

The mechanism of sepsis-induced cardiac dysfunction is believed to be different from that of myocardial ischemia. In sepsis, chemical mediators, such as endotoxins, cytokines, and nitric oxide, cause metabolic abnormalities, mitochondrial dysfunction, and downregulation of β-adrenergic receptors. These factors inhibit the production of ATP, essential for myocardial energy metabolism, resulting in cardiac dysfunction. This review focuses on the metabolic changes in sepsis, particularly in the heart. In addition to managing inflammation, interventions focusing on metabolism may be a new therapeutic strategy for cardiac dysfunction due to sepsis.

## 1. Introduction

Sepsis, defined as life-threatening organ dysfunction caused by a dysregulated host response to infection, is one of the leading causes of death worldwide [1]. Recent research estimates that more than 19.4 million people develop severe sepsis, and 5 million die each year [2]. Therefore, there is an urgent need to understand this disease and find novel strategies for sepsis therapy.

In sepsis-induced organ dysfunction, the most serious complication is cardiac dysfunction, called sepsis-induced cardiomyopathy (SICM) or sepsis-related myocardial dysfunction. The mortality in patients with septic shock is more than 40%, whereas it is 10% in patients without shock [1]. According to some studies, the prevalence of SICM ranges from 10% to 70% in patients with severe sepsis [3,4]. In the state of SICM, a high level of circulating catecholamines downregulates the response of the β-adrenergic receptor (βAR) in cardiomyocytes, blunting the contractile response of cardiomyocytes to catecholamines [5,6,7]. This means that vasopressors to maintain blood pressure do not work for SICM treatment. Although cardiovascular abnormalities in sepsis have been known since the 1950s [8], Parker et al. reported that intrinsic myocardial dysfunction, with an increased volume and a decreased ejection fraction, in patients with sepsis is reversible [9]. Furthermore, these functional changes in the left ventricle were observed 7–10 days from the onset in survivors, whereas they were less profound in non-survivors [9,10]. This suggests that cardiac dysfunction in sepsis is a compensatory mechanism to confer protection from the damage. Following the report, some clinical studies have identified that cardiac dysfunction has reversible systolic and diastolic functions in both the left and right ventricles during sepsis [11,12]. However, recent studies have failed to demonstrate the protective role of cardiac dysfunction [13,14,15]. On the contrary, some clinical researchers suggested that cardiac dysfunction in sepsis is strongly associated with mortality [16,17,18,19]. These different conclusions might result from the complexity of the disease, patient characteristics, methods of diagnosis, and disease access. Notably, the important and interesting characteristics of cardiac dysfunction in sepsis include its reversibility. Despite the advances in our understanding of pathophysiology of sepsis through numerous animal and clinical research studies, the mechanism of SICM remains unclear.

To the best of our knowledge, the mechanism of cardiac dysfunction in sepsis might be different from that of myocardial ischemia. For example, a clinical study demonstrated that coronary flow in patients with septic shock is similar to or higher than that in patients without septic shock [20]. Conversely, many studies have reported that chemical mediators, such as endotoxins, cytokines, and nitric oxide (NO), induced from inducible NO synthase (iNOS) contribute to metabolic disorder, mitochondrial dysfunction, and downregulation of βARs. In particular, the systemic inflammatory response to chemical mediators impairs the ability of cardiac energy metabolism to produce adenosine triphosphate (ATP) in the heart, leading to cardiac dysfunction. In addition, the functional reversibility in SICM might depend on the change in cardiac ATP.

This review focuses on the current knowledge of metabolic alteration in sepsis, especially the dysregulation of cardiac metabolism.

## 2. Energy Metabolism in the Heart

### 2.1. Fatty Acid Metabolism

The main source of cardiac energy is fatty acids (FAs). Almost 70% of ATP production depends on FA oxidation. FAs in serum are transported into cells via cell surface receptors, such as a cluster of differentiation 36 (CD36) or fatty acid-binding protein (FABP) and are converted into long-chain fatty acyl-coenzyme A (fatty acyl CoA) via long-chain acyl CoA synthetases in the cytoplasm. The long-chain fatty acyl CoA is transported into mitochondria via carnitine palmitoyltransferase (CPT)-1, and β-oxidation is performed to produce acetyl-CoA.

Peroxisome proliferator-activated receptors (PPARs) play important roles in cell metabolism. Mammals have three subtypes, PPAR-α, PPAR-δ(β), and PPAR-γ. PPAR-α, highly distributed in the liver and skeletal muscle, promotes the uptake and effective utilization of FAs [21]. PPAR-γ is expressed in adipose tissues and regulates FA synthesis and lipid storage. Interestingly, in the reduction of FAO, PPAR-γ promotes glucose metabolism. PPAR-δ (β) is also known to regulate lipid and glucose metabolism in the heart [22,23].

### 2.2. Glucose Oxidation

Glucose metabolism contributes to ATP production in the decrease of FAO under stress conditions. Glucose transporter 4 (GLUT4) is the most important glucose transporter. The imported glucose is changed into glucose-6-phosphate (G-6-P) and enters various metabolic pathways, such as glycolysis, the pentose phosphate pathway (PPP), and the hexosamine biosynthetic pathway (HBP) [24]. G-6-P is converted to pyruvate in glycolysis, which is utilized to form alanine or lactate in the cytosol. Pyruvate is converted to acetyl-CoA in the mitochondria by pyruvate dehydrogenase and enters the tricarboxylic acid (TCA) cycle for ATP production.

### 2.3. Ketone Metabolism

Ketone bodies consist of acetoacetate, β-hydroxybutyric acid (B-OHB), and acetone, produced from plasma FAs in the liver. In general, ketone bodies are minimally consumed for cardiac energy in non-fasting conditions [25], whereas they contribute largely to energy metabolism in the state of stress or starvation.

Circulating ketone bodies enter cells via monocarboxylate transporters (MCTs) for ketone oxidation. B-OHB is oxidized to acetoacetate by Β-OHB dehydrogenase 1 (BDH1) in mitochondria. Acetoacetate is converted to acetoacetyl-CoA by exchanging CoA between succinyl-CoA and acetoacetate by succinyl-CoA:3 ketoacid-CoA transferase (SCOT). Acetoacetyl-CoA then undergoes a thiolysis reaction of acetyl-CoA acetyltransferase 1 (ACAT1), converting acetoacetyl-CoA into acetyl-CoA, which enters the TCA cycle to produce ATP [26].

### 2.4. Amino Acid Metabolism

The most abundant group of essential amino acids consists of some branched-chain amino acids, including leucine, isoleucine, and valine. Amino acids are transported into cells via amino acid transporters and degraded into acetyl-CoA and succinyl-CoA by amino acid catabolic enzymes, such as branched-chain amino acid transaminases and branched-chain alpha-keto acid dehydrogenase (BCKD). Acetyl-CoA and succinyl-CoA enter the TCA cycle for ATP production [27]. In addition, amino acids activate the mammalian target of rapamycin complex 1 (mTORC1) [28,29].

### 2.5. Mitochondrial Metabolism

Mitochondria produce 95% of the ATP of cells [30]. ATP supply is based on mitochondrial oxidative phosphorylation (OXPHOS), and four electron transport chain complexes (I, II, III, and Ⅳ) and transporters (ubiquinone and cytochrome C) involve the transportation of electrons from the TCA cycle for ATP production. In addition, mitochondria play an important role in calcium signaling. The calcium concentration increase in mitochondria contributes to NADH upregulation [31,32] and ATP production [33].

## 3. Metabolic Alternation in Sepsis

Since Parrillo et al. found a circulating myocardial depressor in the serum of patients with sepsis in 1985 [34], the cardiac depressors have been identified as lipopolysaccharides (LPS), cytokines (interleukin (IL)-1b, tumor necrosis factor-α (TNF-α), and IL-6) [35,36,37], NO synthase (dysregulation of NO synthase (NOS)) [38,39,40], and reactive oxygen species (ROS) [41,42]. In septic conditions, such massive proinflammatory mediators induce systemic and cellular metabolic alternations. In particular, this hypermetabolism in sepsis is characterized by an overall catabolic state, leading to the breakdown of carbohydrate, lipid, and protein stores [43]. In this section, we will outline the different metabolic pathways of the myocardium in sepsis (Figure 1).

### 3.1. Lipid Metabolism in Sepsis

As discussed before, lipid metabolism is the major source of ATP production. During sepsis, most patients are starving and need energy from lipid mobilization and oxidation [44,45]. Increased lipolysis in adipose tissue, the largest energy supplier in the body, converts triglycerides into FAs and glycerol, releasing them into the bloodstream to compensate for energy [46]. Free FAs enter peripheral organs for β oxidation, producing ATP. However, the inflammatory response in sepsis dysregulates various genes associated with lipid metabolism. For example, LPS is reported to reduce PPARα and PGC1α, which regulate the β oxidation pathway [47]. In addition, the dysregulation of CD36 and CPT1 induces a deficiency of FA transportation, leading to impaired FAO [48,49,50]. In particular, decreased CPT1 prevents FAs from entering mitochondria. This situation leads to the accumulation of lipids into the cells, resulting in “lipotoxicity” which causes organ dysfunction.

The association between cardiac lipid accumulation and heart failure in humans was reported long ago. For example, Sharma et al. reported intramyocardial triacylglycerol (TAG) accumulation in patients with heart failure and its transcriptional profile using an animal model of lipotoxicity and contractile dysfunction [51]. Further evidence of lipid accumulation was found in heart samples from patients with severe chronic heart failure underdoing heart transplantation [52].

Lipid accumulation is associated with the cellular apoptotic mechanism [53,54], generation of ROS [55,56], ER stress, and cell death [57], which can lead to heart failure, per se. Although the mechanism of lipid accumulation has not been fully elucidated, lipid accumulation in heart failure underlies a metabolic disorder, especially impaired FAO. We reported a recovery time gap between the expression of CD36 and CPT1 in LPS-induced septic hearts [50]. This time gap leads to the imbalance of FA demand–supply between the cytoplasm and mitochondria and may cause lipid accumulation in the cytoplasm. In addition, the accumulation of fat has been identified in the liver, kidney, and heart muscle in patients with sepsis [58,59,60]. We also observed lipid accumulation in the cardiac tissue of endotoxin model mice. Intriguingly, many lipid droplets were observed around mitochondria in the myocardium 12 h after LPS injection, indicating cardiac dysfunction, but not at 24 h. This suggests a strong relationship between lipid dysregulation and reversible cardiac function.

### 3.2. Glucose Metabolism in Sepsis

Glucose metabolism is also impaired in sepsis. The common characteristic symptom of patients with sepsis is hyperglycemia. Hyperglycemia is a result of the impaired balance between glycolysis and gluconeogenesis, increased insulin resistance, and dysregulation of glucose metabolism. However, hyperglycemia is partly desirable for the immune system. For example, glucose uptake in leukocytes is independent of insulin [61], and activated immune cells depend mainly on energy from aerobic glycolysis [62]. In general, ischemic heart failure increases glycolysis activity to compensate ATP instead of decreased FAO. In the state of hypoxia, aerobic glycolysis synthesizes ATP quickly, despite the ineffectiveness of ATP production. Moreover, glycolysis provides substrates for the synthesis of amino acids, lipids, and nucleotides. At the same time, SICM is reported to decrease not only FAO but also glycolysis. Some animal researchers reported decreased GLUT4 and increased pyruvate dehydrogenase lipoamide kinase isozyme (PDK)-4 in LPS-induced septic hearts [48,49,50]. PDK-4 is an important regulator of FAO and glucose metabolism. PDK-4 negatively regulates GLUT4 to increase FAO [63] and positively regulates PPARα [64].

Intriguingly, Umbarawan et al. reported the accumulation of glycogen in LPS-induced septic hearts [65]. This finding suggests that the septic heart suppresses not only glycolysis and glucose oxidation but also glycogenolysis.

Insulin resistance is also one of the characteristic symptoms of sepsis [66]. In addition, various cytokine effects cause insulin resistance, the release of hyper catecholamine, increased counter-regulatory hormones in plasma, and activation of the sympathetic nervous system [67]. Furthermore, impaired insulin sensitivity of peripheral tissues shifts their metabolism toward lipolysis and proteolysis [68].

### 3.3. Ketone Metabolism

Many researchers have investigated FA and glucose utilization, whereas the knowledge about other nutrients, such as ketone bodies and amino acids, is relatively limited. Fasting conditions, such as sepsis, contribute to systemic hypercatabolism accompanied by increased lipolysis and ketogenesis [69]. Prolonged fasting results in hypoglycemia and promotes ketogenesis in the mitochondria of liver cells. Under limited glucose availability, ketone bodies become an important energy substrate for the brain and skeletal muscles. In addition, this ketogenesis contributes to an important role in biological defense because ketone bodies confer resistance to ROS [70]. Recently, ketone metabolism in heart failure has become known as an alternative fuel that compensates for cardiac energy homeostasis. A mouse model with advanced heart failure induced by TAC and coronary artery ligation demonstrated increased BDH1 expression accomplished by the increased expression of ketone body transporters (MCT1 and MCT2) and increased ^13^C-β-hydroxybutyrate (BHB) utilization based on nuclear magnetic resonance studies [71]. In another report, cardiac-specific BDH1 overexpression in TAC-induced heart failure prevented cardiac remodeling and DNA damage from pressure overloaded stress [71]. Moreover, a clinical study using nondiabetic, nonischemic heart samples of patients at the time of heart transplantation or left ventricular assist device implantation demonstrated an increased abundance of ketogenic β-hydroxybutyryl CoA and increased myocardial utilization of BHB while reducing intermediate and anaplerotic acyl CoA species for the Krebs cycle [72]. Conversely, ketone metabolism in SICM remains elusive. Umbarawan et al. reported that hepatic ketogenesis and cardiac ketolysis would be suppressed by LPS injection in mice lacking fatty acid-binding protein 4 (FABP4) and FABP5 (DKO mice) [65], which are important FA transporters. In their report, the expression of genes related to ketone body production in the liver and 3-oxoacid CoA-transfer (SCOT), required for ketone body degradation in the heart, was significantly decreased in both DKO and wild-type mice. This suggests that ketone metabolism in the heart is also impaired in SICM.

### 3.4. Amino Acid Metabolism in Sepsis

Amino acids are crucial nutrients for cellular homeostasis. According to clinical research, patients lost 13% of total body protein and 70% of the lost proteins from skeletal muscles during sepsis [73]. Sepsis activates proteolysis, the trimming process from proteins into smaller polypeptides and amino acids, in skeletal muscles [74,75]. Although the signaling that induces proteolysis in sepsis is poorly understood, the catabolism of skeletal muscle is necessary to support hepatic gluconeogenesis and protein synthesis. The major reason for increased proteolysis in sepsis is the urgent need for amino acids in the liver to maintain the acute phase response, thus reconstructing energy-abundant molecules [76]. For example, branched-chain amino acids (BCAAs) accept the ketone group of pyruvate and glutamate, which form glutamine and alanine. Glutamine is an important precursor for peptide and protein synthesis, supporting cytokine production. It is also required for purines and pyrimidines to form nucleotide acids for immune cells to increase [77,78,79]. Alanine is transported to the liver and enters the TCA cycle. In addition, essential amino acids are associated with sepsis. Arginine, one of the essential amino acids, is converted into nitric oxide by nitric oxide synthase released from M1 macrophages [80].

Thus, essential amino acids and BCAAs have important roles in maintaining internal homeostasis, including heart function, and they are expressed at high levels in healthy hearts. However, heart failure is known to show abnormal amino acid metabolism. On transcriptome analysis, some genes that regulate the BCAA catabolic pathway were significantly reduced and demonstrated BACC catabolic deficiency in pressure overload-induced mouse failing hearts and human dilated cardiomyopathy [81]. Another report observed dynamic changes in BCKD activity in a mouse heart post myocardial infarction (MI). This study also reported that BCAA catabolic activity is significantly decreased 1 week after MI [82]. Likewise, other studies using the sepsis animal model reported an abnormality of amino acid metabolism. In the mouse cecum ligated puncture (CLP) model, amino acid uptake was reduced by 90% in the heart at 16 h post CLP [82]. Another in vivo study found that metabolic changes in the rat heart of a CLP model showed a 28% decrease in alanine and a 31% decrease in glutamate [83], suggesting that sepsis-induced cardiac dysfunction accompanies impaired amino acid metabolism.

### 3.5. Mitochondrial Metabolism in Sepsis

Mitochondrial dysfunction has been discussed as a critical mechanism of organ failure in sepsis [84,85,86]. Mitochondria are involved in cellular homeostasis systems, such as the cell death pathway, calcium adjustment, redox signaling, and steroidogenesis, including the energy source of cells [87,88,89,90]. During sepsis, the capacity of tissues to utilize oxygen from the blood is reduced despite the normal consumption level of oxygen in tissues [91,92], and excessive ROS are generated from the process of ATP production in mitochondria. Increased ROS in the mitochondrial matrix decreases the efficiency of OXPHOS and leads to reduced ATP synthesis [93,94]. Furthermore, the mechanism of NO production contributes to ROS formation in sepsis. Mitochondria produce NO via mitochondrial NO synthase (mtNOS). This pathway regulates the mitochondrial respiratory chain by inhibition of cytochrome c oxidase [95]. In addition to mtNOS, inducible NOS (iNOS) is generated by various cells and organs, including cardiac tissues [59,96,97]. Although NO has a protective role in maintaining health [98], the excess production of NO generated by iNOS induces vasodilation, decreases cardiac contractility, blunts vascular reactivity to catecholamines, and causes mitochondrial dysfunction.

Cell stress brings about changes in mitochondrial oxygen consumption and OXPHOS. Several studies have reported decreased state 3 respiration, defined as ADP-stimulated respiration, and decreased mitochondrial OXPHOS complexes [50,99] in the septic heart. Decreased state 3 respiration induces a decreased respiratory control ratio, and OXPHOS inactivity decreases the mitochondrial transmembrane potential for the maintenance of energy health [100].

Dysregulation in intracellular calcium homeostasis has also been discussed in sepsis-induced cardiac dysfunction. A literature review showed that cardiac dysfunction in sepsis is associated with the dysregulation of various intracellular Ca^2+^ transporters, which is caused by the inhibition of L-type Ca^2+^ channels, sarcoplasmic (SR) Ca^2+^ ATPase (SERCA), RyR Ca^2+^ leak, and decreased sensitivity in myofilament Ca^2+^ channels [101]. This dysregulation of Ca^2+^ homeostasis causes decreased uptake of Ca^2+^ and increased Ca^2+^ leakage from SR, resulting in mitochondrial Ca^2+^ overload, leading to mitochondrial dysfunction.

In addition, the functional impairment of myocardial mitochondria is accompanied by structural abnormalities, such as mitochondrial deformation, a decrease in the number of mitochondria, and cristae derangement [102,103]. Despite the limited evidence, some clinical studies reported that patients with sepsis have hydropic mitochondria and damage to cristae [102,103]. Furthermore, many studies confirmed mitochondrial abnormalities in experimental animal sepsis models [97,104,105,106,107,108,109]. In contrast, some studies reported mitochondrial dysfunction without abnormalities of mitochondria [110,111,112]. The inconsistent results may depend on the severity of sepsis, the time course of sepsis, and the sepsis model, suggesting that morphological damage to cardiac mitochondria is not a requisite for cardiac function.

## 4. Therapeutic Strategies to Metabolism in SICM

As sepsis is a time-dependent disease, the prompt diagnosis and treatments of sepsis are required. However, even in septic shock, fundamental sepsis therapy is fluid resuscitation with vasopressors and antibiotics. Despite advances in medicine, there are no effective treatments for SICM.

Recently, βAR blockers have attracted unprecedented attention as a treatment for severe sepsis. Originally, βAR blockers were widely prescribed for patients with chronic heart failure caused by ischemia, various types of cardiomyopathy or arrhythmia.

Overstimulating catecholamines in severe sepsis may play a critical role in the onset of SICM by calcium overload and the downregulation of βAR signaling and βARs [7,113,114]. Therefore, the therapeutic strategy to prevent overstimulation of sympathetic nerves could be reasonable. Recent studies have shown that βAR blockade therapy for sepsis reduces the incidence of new arrhythmias, prevents SICM, and even improves mortality [115,116,117,118,119].

The βAR subtypes, β1AR and β2AR, are both expressed in cardiomyocytes, of which β1AR is the main receptor regulating the cardiac contractility response and heart rate [120]. β2AR is associated with glucose metabolism [121], the immune system [122], and inflammatory mediators [123,124]. Collectively, βAR blockers affect the metabolism, immunology, and cardiac alternation induced by sepsis [125]. Notably, βAR blockers are an attractive strategy for sepsis. β1AR blockers suppress excess catecholamine release, which prevents the hypercatabolic state in sepsis. Moreover, the block of β1AR decreases heart rate and prolongs the filling time of the left ventricle, thus increasing the stroke volume and maintaining cardiac output. Therefore, βAR may add a novel line of conventional therapy for SICM.

Currently, we are focused on the effects of β3AR on cardiac metabolism in SICM and have reported that β3AR blockade improved cardiac dysfunction and mortality in mice with LPS-induced heart failure [50]. Most β3AR exists in fat cells and is associated with lipolysis, whereas there are a few β3ARs in the heart. Interestingly, contrary to other βAR subtypes, β3AR increases in ischemic heart failure [126], and β3AR stimulation improves cardiac function post MI via NOS regulation [127,128]. Against our expectation, however, blockade of β3AR significantly improved cardiac ATP and mortality, whereas β3AR had the opposite result. Our findings demonstrated that (1) β3AR is increased in septic hearts; (2) β3AR blockade preserves cardiac ATP through the improvement of FAO; (3) β3AR blockade prevents lipid accumulation in the myocardium; and (4) β3AR regulates iNOS. This research suggests that blockade of β3AR improves FAO and ATP production in hearts by suppressing sepsis-induced iNOS generation. However, many researchers argue that B3AR activation is useful for improving cardiovascular pathogenesis [129]. As the pathogenesis of sepsis involves changes in immunity and metabolism over time, it may be important to identify the phases of the “metabolic switch.”

In another study, Drosatos et al. reported that PPAR-γ activation improved cardiac dysfunction and mortality in LPS-induced septic mice by preserving cardiac FAO and ATP. In contrast, PPAR-α activation failed [49]. Although cardiac metabolism in sepsis is complex, the metabolic approach can provide a new therapeutic strategy for SICM.

## 5. Conclusions

This review focuses on cardiac metabolism in sepsis. Sepsis is one of the leading causes of death, and SICM is associated with a worse outcome of sepsis. Although the mechanism of SICM has not been fully elucidated, many laboratory experiments and clinical data have accumulated evidence that ATP depletion by metabolic disorders contributes to SICM. In particular, the cardiac metabolism in sepsis is different from that in the ischemic heart, switching the energy pathway from FAO to glycolysis. As discussed above, SICM may not rely on any metabolic pathways because strong inflammation makes all ATP pathways dysregulated. Moreover, the imbalance between hypercatabolism to compensate energy and dysregulation of metabolic genes leads to detrimental effects, such as lipid accumulation. Therefore, when treating SICM, the focus should not only be on managing inflammation, but also on intervening in cardiac metabolism.

## Figures and Tables

**Figure 1 metabolites-11-00846-f001:**
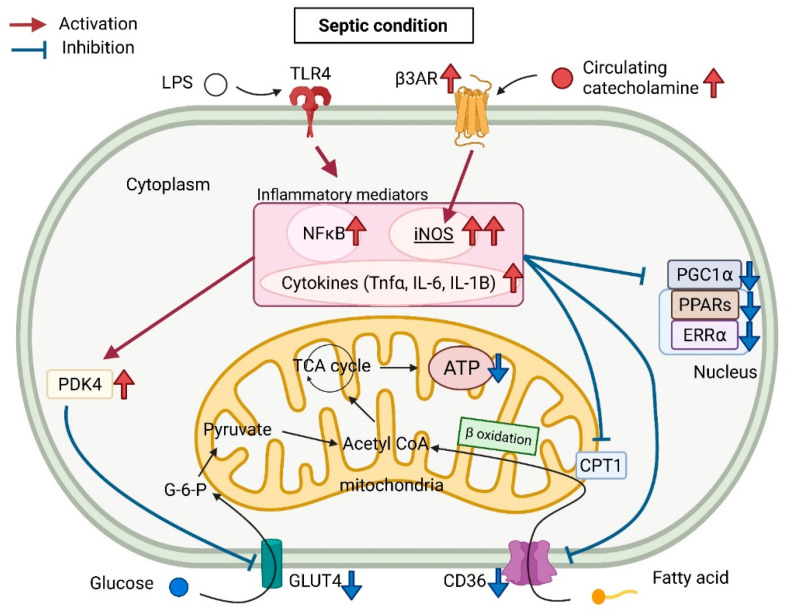
Schematic representation of myocardial metabolism in a septic condition.

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
