# Peer review of "Cardiac Metabolism in Sepsis"

_metabolites, 2021, doi:10.3390/metabo11120846_

Round 1

Reviewer 1 Report

The review by Kawaguchi and Okada discusses mechanisms of sepsis-induced cardiac dysfunction and proposes that, in addition to modulating inflammation, interventions targeting metabolism may be a new therapeutic strategy.

General Comments

The review adds to a growing literature on sepsis-induced-myocardial dysfunction (SIMD) (see e.g. Lin Y, Xu Y, Thay Z.  Inflammation 2020, and many other reports).  It is well written and my suggestions relate to the style and a few facts that still need clarifications.

Specific Points

  1. There are already many brilliant reviews on cardiac metabolism in the literature, and the section on “Energy metabolism in the Heart” can be shortened by 50%.
  2. Line 155: Please define “lipotoxicity”.
  3. A major mistake I discovered is in line 193: “glycogenesis” should be “glycogenolysis”.
  4. Another major point relates to Ref. 72 in line 218: ß-OH reducing intermediate and anaplerotic acyl-CoA species? Please define “anaplerotic acyl-CoA species.

Other Points

  1. Line 337: The sentence beginning with “Although………” is very general.  Can you be more specific on which metabolic approach can provide a new therapeutic strategy for SICM?

Author Response

Thank you for your many constructive suggestions.

We have revised it according to your suggestions. We would be grateful if you could check it and give us further advice.

Reviewer 2 Report

This is a comprehensive review on the metabolic changes of cardiomyocytes during sepsis. The manuscript is well organised and cited most important reference. However, the Figure 1 is too simple and not well presented. It would be better to make a more comprehensive diagram to cover most important points in text with a good art work.

Author Response

 Thanks for the favorable peer review. Also, sorry for the confusing figure.

                  We have revised the figure after your suggestion, and we would appreciate if you could check it and advise us.
